# SWE-rebench V2: Language-Agnostic SWE Task Collection at Scale

**Ibragim Badertdinov** [1]  **Maksim Nekrashevich** [1]  **Anton Shevtsov** [1]  **Alexander Golubev** [1]

## Abstract

Software engineering agents (SWE) are improving rapidly, with recent gains largely driven by reinforcement learning (RL). However, RL training is constrained by the scarcity of large-scale task collections with reproducible execution environments and reliable test suites. Although a growing number of benchmarks have emerged, datasets suitable for training remain limited in scale and diversity or often target a limited set of high-resource language ecosystems. We introduce SWE-rebench V2, a language-agnostic automated pipeline for harvesting executable real-world SWE tasks and constructing RL training environments at scale. The pipeline synthesizes repository-specific installation and test procedures via an interactive setup agent, and filters unsound instances using an ensemble of LLM judges, validated against human-verified SWE-bench annotations. Using this pipeline, we construct a dataset of 32,079 tasks spanning 20 languages and 3,617 repositories, with pre-built images for reproducible execution. To further scale training data, we additionally release 120,000+ tasks with installation instructions, fail-to-pass tests and rich metadata, where the problem statement is generated based on the original pull request description. We validate the collected instances through a diagnostic study that covers a subset of tasks in five programming languages across seven popular models, and provide instance-level metadata that flags common confounders such as overly restrictive tests and underspecified descriptions. We release the datasets, the collection and execution code, and associated artifacts to enable large-scale training of SWE agents across diverse languages and repositories.

## 1. Introduction

Large language models (LLMs) increasingly power software engineering (SWE) agents that operate directly on real repositories by proposing patches and iterating with tool feedback. A natural way to evaluate these agents is repository-level issue resolution, where an agent fixes a real repository issue and correctness is verified by executing the project test suite. Benchmarks in this style, beginning with SWE-bench (Jimenez et al., 2024) and followed by SWE-bench Verified (Chowdhury et al., 2024) and harder variants, have become a standard basis for evaluating SWE agents. Training on such executable environments, typically via reinforcement learning with test-based rewards and tool feedback, has already been shown to improve LLM code-agentic capabilities (Luo et al., 2025; Sun et al., 2025; Wei et al., 2025; Golubev et al., 2025; Yang et al., 2025b; Wang et al., 2025a).

Despite this progress, training autonomous SWE agents is bottlenecked by the availability of executable task environments. Reinforcement learning (RL) and other interactive learning paradigms require large numbers of tasks with stable reward signals; in repository-level settings, this stability depends on (i) correct dependency installation, (ii) reliable and reproducible test execution, and (iii) alignment between the natural-language specification and the test oracle. Constructing such environments is costly even within a single ecosystem, and scales poorly across programming languages due to heterogeneous build systems, dependency managers, and test runners.

Recent work has expanded evaluation beyond Python through multilingual benchmarks such as Multi-SWE-bench (Zan et al., 2025) and SWE-PolyBench (Rashid et al., 2025), demonstrating that language diversity materially changes agent behavior and task difficulty. At the same time, these datasets highlight a persistent tension: achieving high-confidence executable instances often requires substantial manual verification, limiting scale and reducing utility as a training substrate. In parallel, automated pipelines such as SWE-rebench (Badertdinov et al., 2025), SetUpAgent (Vergopoulos et al., 2025), SWE-bench-Live (Zhang et al., 2025a), SWE-Factory (Guo et al., 2026), and SWE-Bench++ (Wang et al., 2025b) have improved scalability by automating environment setup and instance vali-

[1]Nebius, Amsterdam, Netherlands. Correspondence to: Alexander Golubev <alex_golubev@nebius.com>.

*Proceedings of the 43rd International Conference on Machine Learning*, Seoul, South Korea. PMLR 306, 2026. Copyright 2026 by the author(s).

dation. However, two obstacles remain for training-focused research. First, it is unclear what it means for a construction pipeline to be language-agnostic in practice: many systems are evaluated on a small number of ecosystems, leaving open questions about robustness to long-tail toolchains and repository conventions. Second, most released resources are evaluation-first rather than for large-scale interactive learning, and often omit training-oriented artifacts needed for reproducible RL at scale, such as pre-built environments and fine-grained instance-level diagnostics.

This paper introduces SWE-rebench V2, a language-agnostic pipeline designed to generate interactive training environments at scale. By language-agnostic, we mean that the same end-to-end construction workflow applies across languages, while relying on a small set of reusable language-specific templates (base images, runners, and parsers). SWE-rebench V2 mines pull request histories (and linked issues when available), constructs containerized environments for diverse ecosystems, extracts fail-to-pass tests, and applies automated quality assessment to filter ambiguous or invalid tasks without per-instance human verification.

Our contributions are as follows:

- We introduce a language-agnostic construction funnel that combines interactive environment synthesis with automated oracle extraction and quality filtering, and we quantify the yield and failure modes of each stage across ecosystems.

- We release 32,000+ containerized SWE tasks from 3,600+ repositories across 20 programming languages, with executable environments and pre-built images[2].

- We additionally release 120,000+ SWE tasks with installation/test recipes and metadata, using pull request descriptions as problem statements to enable substantially larger-scale learning across diverse development activities and avoiding reliance on issue linkage[3].

- We provide instance-level diagnostic metadata (e.g., external dependencies, test brittleness, underspecification) derived from analysis of hundreds of tasks and thousands of trajectories across seven frontier models, enabling stratified filtering, curriculum design, and controlled analyses in large-scale training.

- We perform ablations on setup synthesis by comparing a non-interactive setup pipeline to an interactive setup agents with different models and retry budgets, and on automatic issue clarity filtering by varying prompts, models, and ensembling against human annotations.

---

[2]Available at: HF: nebius/SWE-rebench-V2.

[3]Available at: HF: nebius/SWE-rebench-V2-PRs.

## 2. Related Work

**Repository-level issue resolution benchmarks.** SWE-bench introduced execution-based evaluation on real GitHub issue–pull request pairs and established fail-to-pass testing as the primary oracle. Subsequent releases emphasize higher-confidence evaluation through human verification and/or harder task distributions (e.g., SWE-bench Verified). Several efforts further address benchmark aging and contamination via continual refresh from recent issues, e.g., SWE-bench-Live (Zhang et al., 2025a), SWE-rebench (Badertdinov et al., 2025). SWE-bench Pro (Deng et al., 2025) pushes difficulty further through careful data selection and a human-centered augmentation workflow that adds structured requirements and explicit interface specifications, reducing false negatives where models produce functionally correct fixes but misname symbols expected by tests. Multilingual evaluation benchmarks extend this setting beyond Python. Multi-SWE-bench covers seven languages with expert annotation and manual verification, and SWE-PolyBench provides a curated multi-language benchmark spanning Python, Java, JavaScript, and TypeScript with an automated evaluation harness. SWE-bench-java (Zan et al., 2024) and SWE-Sharp (Mhatre et al., 2025) explicitly target other popular languages. These datasets demonstrate the importance of language diversity for evaluation, but their construction costs and scales are typically aligned with benchmarking rather than large-scale training.

**Automated instance construction and environment setup.** A growing line of work targets automation of the instance construction funnel, especially accurate environment setup and test execution. SWE-rebench introduced a fully automated pipeline for mining and validating large-scale executable tasks in Python. SetUpAgent automates dependency setup, test execution, and result parsing to generate broader (largely Python-centric) benchmarks from less-popular repositories. SWE-Factory proposes an automated pipeline spanning four languages, including a multi-agent builder for environment construction and an exit-code-based grading scheme for oracle validation. SWE-Bench++ scales multilingual instance generation via staged pipelines that include environment synthesis, test oracle extraction, and quality assurance, releasing a public benchmark subset for evaluation. RepoForge (Chen et al., 2025) emphasizes end-to-end curation and infrastructure efficiency, combining automated environment generation with storage reduction and distributed evaluation.

SWE-rebench V2 builds on these efforts by targeting *language breadth under a single executable contract*, by releasing *training-oriented artifacts* for reproducible interactive learning, including pre-built environments and by rigorous instance-level diagnostics. Our experiments quantify how setup strategy and quality filtering interact across diverse

build ecosystems.

Many issue-resolution benchmarks use the issue text as the primary specification and therefore depend on reliable issue–PR linkage. SWE-rebench V2 supports both issue-based specifications and a pull-request-derived formulation in which the problem statement is taken directly from the PR description, enabling a substantially larger recipe-scale corpus for learning while keeping a separate fully containerized release for reproducible execution.

**Training environments and task corpora for agent learning.** Beyond evaluation, several datasets explicitly frame issue resolution tasks as training environments. SWE-Gym (Pan et al., 2025) provides a Python environment for training agents with executable runtimes and releases trajectory data for learning and verification. Multi-SWE-bench also introduces Multi-SWE-RL as an initial RL-oriented multilingual task set. These efforts motivate learning-oriented datasets, but they remain limited in language coverage or scale relative to the training.

**Automated labeling and instance quality assessment.** Instance quality is increasingly recognized as a bottleneck for both evaluation and learning. SPICE (Oliva et al., 2025) proposes automated labeling for attributes such as issue clarity and test coverage using multi-pass consensus, and reports agreement with human-verified SWE-bench annotations. SWE-rebench V2 integrates automated quality assessment directly into the construction funnel, calibrates filter behavior against human-verified data, and pairs filtering with a diagnostic analysis of failure modes that distinguishes model limitations from environment pathologies (e.g., flaky tests, external dependencies), producing actionable instance-level metadata for downstream filtering and controlled analyses.

**Synthetic and test-driven data generation.** Several works scale training data via synthetic instance generation. SWE-smith (Yang et al., 2025a) generates task instances by inducing test failures in Python repositories, while SWE-Flow (Zhang et al., 2025b) constructs test-driven tasks by synthesizing partial codebases, tests, and modifications. Synthetic data can provide scale and controlled curricula, but may not capture the noise, ambiguity, and tooling variability present in real issue reports. SWE-rebench V2 therefore focuses on harvesting real issue-resolution histories while providing optional enrichments and diagnostics that improve usability for learning without altering the underlying task distribution.

## 3. Pipeline

We develop a language-agnostic automated pipeline for collecting executable software engineering tasks with test-

based verification at scale. The pipeline enables the autonomous construction of 32,000+ executable tasks spanning 20 programming languages and 3,600+ repositories. Our methodology follows the standard execution-based dataset construction process: mine historical changes, build reproducible environments, and verify solutions via running the tests. While the pipeline operates across diverse ecosystems, it does not eliminate all language-specific components. Instead, it enforces a unified construction workflow in which language-specific artifacts (e.g., base images, runners, and parsers) are reused and generated automatically, enabling scaling to new languages without manual engineering.

Concretely, the pipeline is designed to:

1. Handle heterogeneous build systems and test runners across diverse ecosystems, including long-tail languages with non-standard toolchains.

2. Infer installation and test procedures once per repository and reuse them across all tasks mined from that repository.

3. Pair each task with rich diagnostic metadata to identify confounding factors such as flaky tests or underspecified specifications, enabling controlled training and fine-grained evaluation.

The pipeline has five stages:

1. **Preliminary Data Collection:** Mining and filtering candidate PRs from global GitHub activity.

2. **Setup Synthesis:** Deploying an interactive agent to autonomously infer repository-level installation and testing scripts.

3. **Execution-based Validation:** Validating environments through dual-pass execution (pre- and post-fix) to extract test oracles.

4. **Filtering By Issue Clarity:** Removing underspecified tasks and excluding instances with potentially sensitive information.

5. **Metadata Enrichment:** Tagging instances with diagnostic features to enable flexible selection.

### 3.1. Preliminary Data Collection

We aggregate public activity from GitHub Archive, join issue and pull request metadata, and clone repositories to extract patches from commit histories. Using GitHub Archive as a primary source, we collect issue descriptions, PR discussions, commit SHAs, and repository-level attributes such as licenses and primary languages. To bypass GitHub API rate

limits and enable large-scale processing, we clone repositories in distributed map-reduce jobs and extract pull request patches directly from local git histories, avoiding per-instance API requests.

Repositories are filtered based on language and the number of closed issues to optimize the compute-to-yield ratio. We then link issues to pull requests that reference resolving them in the PR title or description, and apply instance-level filters to keep candidates that (i) belong to repositories with permissive licenses, (ii) correspond to resolved issues and merged pull requests. To enable automatic verification, we require pull requests that introduce or modify tests.

For each selected pull request, we split the overall diff into a solution patch (non-test files) and a test patch (test files only). Test files are identified using the regular expression `(?i)(test(?:ing|s)?|e2e)`. Because some languages permit inline tests within production files, we apply a rubric during metadata enrichment to detect and flag if test logic is contained in the file with solution patch. For high-resource ecosystems (e.g., Python, Java, Go), we apply strict filters (minimum 25 stars and 15 closed issues), since task yield is typically highly skewed and a small subset of core repositories accounts for the majority of verifiable tasks. Appendix A.1 shows that this strategy retains about 20% of repositories while covering roughly 80% of tasks, reducing the number of repositories requiring setup synthesis by a factor of five. For long-tail languages, we relax these thresholds (10 stars and 1 closed issue) to preserve diversity and avoid discarding most of the already limited repository pool. After filtering, this stage yields approximately 21,000 repositories and 580,000 task instances.

### 3.2. Setup Synthesis

To execute each task and verify candidate fixes, we must construct a correct project environment where tests run deterministically. We represent each environment as a Docker image containing the repository source and all required dependencies, so tests can run without network-dependent installs at runtime.

For each language, we pre-build base Docker images that include the runtime and a small core set of common tooling. We generate a base Dockerfile for each language from a template using Qwen3-Coder-480B-A35B-Instruct model (Team, 2025). For large ecosystems, we provide multiple base images corresponding to commonly used major toolchain versions to support both legacy and modern repositories (e.g., separate Java base images with JDK 11, JDK 17, and JDK 21). An example base Dockerfile is shown in Appendix A.2. The prompt used for the base Dockerfile generator is shown in Appendix A.3.1. We infer setup once per repository by synthesizing installation and test procedures on a representative snapshot corresponding to the latest task

we mined from that repository (latest by PR merge time among the repository's collected tasks), and then reuse this setup across all tasks from the repository.

The interactive setup agent operates in a closed-loop debugging cycle: it inspects the codebase, attempts dependency installation, and iteratively refines scripts based on observed build errors and test failures, with success defined as running the project test suite without infrastructure or dependency failures. We employ the mini-SWE-agent v1.14.4 (Yang et al., 2024) scaffold with Qwen3-Coder-480B-A35B-Instruct as the underlying model to generate installation and test scripts. This choice is based on ablation studies comparing different setup strategies and underlying models; detailed results are reported in Section 4.1. When supported, we enforce structured test reports (e.g., JUnit XML) and prefer standard package managers to enable unified parsing and robust error handling. In JVM-based languages such as Scala, the ordering and naming of tests in stdout can vary across runs, making output-based parsing unreliable, whereas structured XML reports provide a stable and unambiguous report. For compiled languages such as C/C++, where applying a patch invalidates previously built artifacts, the agent explicitly inserts recompilation commands before running the test suite. Concretely, after applying the solution patch, the agent runs an explicit rebuild step (e.g., clean and compile) so that the subsequent test run executes binaries produced from the patched sources rather than stale artifacts from the pre-patch build. The prompt used for the setup agent is shown in Appendix A.3.2.

After the agent produces an `install_config`, we re-run installation and the test suite to verify that the synthesized procedure is reproducible. Each task must also have a log parser to convert raw test output into structured results. We bootstrap this parser from logs of a task whose tests are executed successfully without infrastructure errors, and then apply it to other tasks from the same repository. From a batch of raw test logs, we use Qwen3-Coder-480B-A35B-Instruct with a fixed prompt provided in Appendix A.3.3 to synthesize a repository-specific parser that maps output into a normalized schema (pass, fail, error, and test identifiers). If parsing fails on remaining logs, we regenerate parsers from new successful traces up to five iterations. An example of a generated parser is provided in Appendix A.3.5. In most ecosystems, especially those that emit XML reports or have a standard test runner, one or two parsers are sufficient per repository; however, some ecosystems require more specialized parsing logic (e.g., C/C++ projects using different runners such as CTest, GoogleTest, or Catch2 often produce heterogeneous stdout formats). After each test run, we parse execution logs into structured outcomes.

## 3.3. Execution-based Validation

We use multi-stage Docker builds to separate reusable base layers from repository-specific installation and testing. A pre-built base image provides the language runtime and common tooling, improving cache reuse and reducing build time and final image size.

We then apply the test patch and run the full test suite. Next, we additionally apply the solution patch and rerun the full test suite. This produces paired execution traces before and after the fix for each candidate instance. We always run the full project test suite, rather than selecting task-specific subsets. Full-suite execution increases coverage and helps to detect unintended side effects. We retain only instances with at least one fail-to-pass test, ensuring a non-trivial executable signal for learning and evaluation. To reduce noise from flaky or nondeterministic tests, we rerun each candidate task three times and retain only instances whose structured test outcomes remain unchanged across runs.

## 3.4. Filtering by Issue Clarity

We perform preliminary automated filtering of issue text to remove tasks that are likely underspecified. We follow the SWE-bench Verified setup and adapt its annotation rubric into an LLM-friendly prompt. We score each issue with three independent LLM judges (gpt-oss-120b (OpenAI, 2025), GLM-4.7 (Team et al., 2025), and DeepSeek-V3.2 (DeepSeek-AI et al., 2025)) and retain an instance only when all three judges rate the specification as adequate for implementation. An ablation study analyzing this filtering strategy and different model combinations is presented in Section 4.2.

## 3.5. Metadata Enrichment

Based on the analysis of seven frontier model runs described in the Section 4.3, we develop a meta-prompt that automatically annotates each task with potential limitations and characteristic features. These metadata allow researchers to select subsets of tasks, for example, by estimated difficulty or by task type.

The annotation is performed using the gpt-oss-120b model. In addition, for each task we automatically generate auxiliary interfaces extracted from the patches that are explicitly exercised in the test suite. These interfaces consist of method or class names together with their signatures and a brief description. The prompt used for interface generation is provided in Appendix A.3.7.

## 3.6. PR-based Task Expansion

After constructing the final set of issue-based tasks, we expand the dataset by incorporating pull requests that are not directly linked to issues. For this, we consider repositories where executable tasks were successfully collected and reuse the previously synthesized installation and test instructions, joining them with standard pull requests from the same repositories.

For these candidates, we generate a synthetic problem description conditioned on the pull request description and the corresponding patch, rather than relying on the raw PR text. To mitigate potential solution leakage into the task description, we refine the prompt and apply additional post-processing to detect and filter suspicious cases. The prompt for generating problem statements is provided in Appendix A.3.4. We release this PR-derived collection as an additional training resource, accompanied by metadata signals to support subset selection. To quantify leakage in PR-derived tasks, we evaluated 509 PR-derived tasks constructed from SWE-bench Pro pull requests. An LLM judge compared each generated statement against the reference problem statement, requirements, and interfaces, and rated whether the statement introduced implementation details beyond the reference. Under this protocol, 392/509 tasks (77.0%) were clean, 117/509 (23.0%) contained some leakage, and only 12/509 (2.4%) contained explicit solution leakage. We therefore release the PR-derived split as a larger but lower-confidence training resource, accompanied by metadata to support filtering.

*Table 1.* PR and repository filtering stages

| Stage | PRs | Repos |
|---|---|---|
| PRs | 29,511,758 | 145,306 |
| PRs with tests | 8,593,722 | 101,958 |
| PR linked with issue and test | 805,598 | 50,797 |
| Repo based filtering | 583,809 | 21,692 |
| Successful tasks w/ F2P | 41,349 | 4,006 |
| Issue text based filtering | 33,049 | 3,701 |
| Stable across 3 validation runs | 32,079 | 3,617 |

## 3.7. Pipeline Funnel

Table 1 summarizes the end-to-end filtering funnel from raw GitHub pull requests to executable tasks. Starting from 29.5M PRs across 145k repositories, the first major reduction comes from requiring tests, which removes a large fraction of PRs that cannot be directly used with a test-based oracle. A second strong bottleneck is issue linkage: restricting to PRs that are both linked to issues and contain tests reduces the candidate pool by nearly an order of magnitude, motivating our PR-based task expansion where the problem statement is generated from the pull request description and patch.

Repository-level filtering is designed to reduce the number of repositories that require expensive setup synthesis while retaining most tasks. We apply stricter thresholds for

high-resource languages to keep setup costs manageable, and relaxed thresholds for long-tail languages to preserve diversity. Even with an interactive setup agent, only around 20% of repositories succeed with a single setup attempt per repository, suggesting headroom from additional retries and from considering multiple repository states (e.g., different commits or toolchain versions) where setup procedures may differ.

The final dataset contains 32,079 tasks spanning 2014–2025. The median task modifies 3 files and 34 lines and is medium difficulty, however the distribution is heavy-tailed (90th percentile at 9 files / 181 lines) including a lot more challenging tasks. The benchmark is multi-language (20 languages), led by Python (21.6%) and Go (20.6%), followed by JS/TS/Rust. Tasks cover up to 12 PR categories, including bug fixes, regressions, documentation, dev-ops, performance, integration, UI/UX, and security. More detailed statistics can be seen in the Appendix B.

The full construction run required substantial but tractable compute. The dependency-installation agent processed 21,692 repositories, averaging 24.67 interaction turns per repository, for approximately 535K total model API calls. Each repository-level trajectory used roughly 214K input tokens and 966 output tokens on average; at representative commercial API pricing this corresponds to about $0.0873 per repository run, or roughly $1.9K total for setup inference. Task-level validation required 583,809 Docker builds, with an average build time of 2.71 minutes, totaling approximately 26,400 job-hours. The final Docker artifacts for the 32,079 retained environments occupy 26.36 TiB.

## 4. Experiments and Details

### 4.1. Setup Synthesis

The setup synthesis stage critically impacts the final task yield, as successful environment configuration is a prerequisite for validation. Prior approaches to setup automation vary: early benchmarks often relied on manual environment preparation, whereas recent pipelines increasingly use LLM-based or agentic systems. A non-interactive approach follows a fixed sequence: an LLM analyzes a predefined set of repository files to generate installation instructions. While effective in structured ecosystems like Python, this approach is less suitable for broad multilingual coverage, where a unified, interactive agent is more robust.

We implemented and compared several variants of this stage. The first is a non-interactive pipeline with three fixed steps: (1) analyzing a file shortlist to identify setup instructions, (2) generating installation and test commands, and (3) refining instructions based on error logs. The non-interactive baseline uses Qwen3-Coder-480B-A35B-Instruct with the same base images and task inputs as the interactive agent.

The second variant is a fully interactive agent based on mini-SWE-agent, instantiated with different underlying models. We conduct an ablation study to evaluate the effect of (i) interactivity (agent vs. non-interactive pipeline), (ii) model choice, (iii) the number of setup attempts (runs), and (iv) context length. We run experiments on a subset of 103 tasks, each from a unique repository, sampled from SWE-bench, SWE-bench-multilingual, and Multi-SWE-Bench, covering ten different languages in total, with distribution, reported in Table 9. For each configuration, we perform 10 independent runs on this subset. To establish a ground truth, we convert the manually written setup instructions provided with these tasks into our pipeline format and verify their correctness and reproducibility on our base Docker images. The automated setup systems then attempt to replicate this process on top of the same base images. Success is measured by comparing the fail-to-pass test set produced by the automated setup with the reference set from the validated manual setup; a perfect match indicates a successful configuration.

Results are reported in Table 2 and suggest several conclusions. First, while long context can help on some repositories, 32k tokens appear sufficient for most projects. Trivial setups are resolved quickly regardless of context, and longer context can increase the risk of the agent getting stuck in loops or focusing on irrelevant details. Second, interactive systems consistently outperform non-interactive ones: for example, an interactive agent using Qwen3-Coder-30B-A3B-Instruct can exceed a non-interactive pipeline even when the latter is driven by Qwen3-Coder-480B-A35B-Instruct. Third, increasing the number of setup attempts improves success probability: moving from one to ten runs substantially increases the fraction of installed repositories, in some settings approaching a twofold improvement.

Common setup failures include selecting an incompatible toolchain version, missing native or system dependencies, incorrect monorepo root selection, private registry or credential requirements, heterogeneous test-runner output, and premature convergence after an initially plausible but incomplete setup.

For our main pipeline, we use a single setup run as a cost-yield trade-off to maximize task throughput for large-scale data collection. However, this stage is configurable: the number of runs can be increased for specific languages or repositories to maximize task yield when additional compute is available.

### 4.2. Filtering by Issue Clarity

To choose an effective configuration for preliminary filtering based on issue descriptions, we conduct ablations over prompts, models, and ensembling strategies.

For testing, we use the SWE-bench Verified annotation

*Table 2.* Installation agent results (pass@k). Qwen3-480B denotes Qwen3-Coder-480B-A35B-Instruct; Qwen3-30B denotes Qwen3-Coder-30B-A3B-Instruct.

| Model | pass@1 | pass@3 | pass@5 | pass@10 |
|---|---|---|---|---|
| Non-interactive setup | 12.1 | 14.2 | 14.9 | 15.7 |
| mini-SWE-agent (Qwen3-30B, 32k) | 17.4 | 30.5 | 36.9 | 46.1 |
| mini-SWE-agent (DeepSeek-V3.2, 32k) | 20.3 | 37.3 | 46.6 | 59.8 |
| mini-SWE-agent (Qwen3-480B, 32k) | 25.8 | 42.4 | 50.0 | 58.8 |
| mini-SWE-agent (Qwen3-480B, 64k) | 26.8 | 43.1 | 49.2 | 55.9 |
| mini-SWE-agent (Qwen3-480B, 128k) | **27.1** | **44.4** | **52.2** | **62.7** |

dataset. It consists of 1699 instances from SWE-Bench each scored by 3 human annotators by multiple criteria, including: (1) *well-specified* – how well the issue text defines the problem, (2) *valid evaluation criteria* – how well the test patch validates the solution, and (3) *difficulty* – time estimate to come up with a solution.

For the well-specified criterion, the issue text is assigned a score from 0 to 3 with 0,1 corresponding to well-specified and 2,3 to underspecified issues. The final label is the maximum of three scores. We compare the annotation produced by LLM to this labeling and ablate following components:

- **Prompt** We test several modifications to the annotation prompt. We consider two modifications to the baseline prompt. First, we use GPT 5.2 to rewrite the instruction with additional valuable advice. We refer to this modification as Verified+. Second, along with issue description, we provide the model with patch and test patch. We refer to this as Verified-E. Additionally, we include the prompt from SPICE into our comparison. As shown in Table 3, Verified+ achieves the best F1 score, while Verified-E achieves the highest precision. Since precision is particularly important for our filtering stage, we use the Verified-E configuration throughout the pipeline. The full Verified-E prompt is provided in Appendix A.3.8.

- **Model** We test the performance of popular open and proprietary LLMs. For each model, we use the default parameters. Across individual judges, gpt-oss-120b provides the best balanced performance, while several models trade recall for higher precision. The full list of models can be found in Table 4.

- **Ensembling** We compare various ensembling strategies. First, we consider aggregation of annotations by the same model. We also consider two aggregations of the scores from gpt-oss-120b, GLM 4.7 and DeepSeek v3.2: average and consensus. Table 5 demonstrates that ensembling by averaging scores improves robustness and yields the best overall F1, whereas strict consensus is beneficial when precision is prioritized.

*Table 3.* Issue clarity filtering results across prompt configurations.

| PROMPT | ACC | PREC | REC | F1 |
|---|---|---|---|---|
| REBENCH V1 | 0.67 | 0.76 | 0.22 | 0.34 |
| SPICE | 0.66 | 0.59 | 0.34 | 0.43 |
| VERIFIED | 0.68 | 0.75 | 0.24 | 0.36 |
| VERIFIED+ | **0.69** | 0.66 | **0.40** | **0.50** |
| VERIFIED-E | 0.65 | **0.83** | 0.10 | 0.17 |

*Table 4.* Issue clarity filtering results across LLMs. All models are run with Verified prompt.

| MODEL | ACC | PREC | REC | F1 |
|---|---|---|---|---|
| GPT-OSS-120B | **0.68** | 0.75 | **0.24** | **0.36** |
| GPT-OSS-120B (HIGH) | 0.66 | 0.81 | 0.16 | 0.26 |
| DEEPSEEK V3.2 | 0.66 | 0.76 | 0.18 | 0.29 |
| GLM 4.7 | 0.64 | 0.81 | 0.10 | 0.18 |
| MINIMAX M2.1 | 0.59 | 0.59 | 0.16 | 0.25 |
| GPT 5.2 | 0.67 | 0.80 | 0.18 | 0.30 |
| GEMINI 3 PRO | 0.57 | **0.92** | 0.05 | 0.10 |
| CLAUDE OPUS-4.5 | 0.67 | 0.83 | 0.16 | 0.27 |

### 4.3. Task Analysis

To assess how our dataset supports large-scale training and to characterize imperfections inherent to the pipeline, we conducted a comprehensive analysis of environmental pathologies. We utilized a subset of 300 tasks (60 randomly selected per language: Python, JavaScript, Go, Rust, Scala) and evaluated seven frontier models: DeepSeek-V3.2, Gemini3-Flash, GLM-4.7, GPT-5.2 medium, gpt-oss-120b, MiniMax-M2.1, and Claude Opus-4.5. We employed mini-SWE-agent with default generation parameters for each model.

Each model performed three independent runs per task. Table 6 summarizes the pass rates, establishing a baseline for state-of-the-art models. Detailed per-language results, including confidence intervals, are provided in Appendix C.1.

By analyzing execution trajectories from these runs, we identified specific failure modes and uncovered systematic issues in task formulation that impact training signal validity:

**Test Suite Coupling.** We discovered cases where models correctly fixed the target issue but failed due to regressions in unrelated code paths. While this suggests the tests validate a narrow set of implementation paths, it is not necessarily a defect; the failure often correctly indicates valid regressions caught by the existing pass-to-pass (P2P) test suite, providing a signal for regression avoidance.

**Implicit Naming Requirements.** Tests often expect specific implementation details not specified in the problem statement. Consequently, a model implementing the literal

*Table 5.* Issue clarity filtering results across ensembles. All models are run with Rebench V1 prompt.

| SETUP | ACC | PREC | REC | F1 |
|---|---|---|---|---|
| SINGLE MODEL | 0.67 | 0.76 | 0.22 | 0.34 |
| MIXED (AVG) | **0.69** | 0.73 | **0.31** | **0.43** |
| MIXED (CONSENSUS) | 0.64 | **0.88** | 0.06 | 0.11 |

specification would fail despite correct logic. This issue can be mitigated by injecting hints about the naming conventions expected by the test suite into the problem description, as described in Section 3.5.

**External Dependencies.** Some tasks reference external URLs in their problem statements, such as GitHub issues, API documentation, or design documents. These resources may be inaccessible to the agent during evaluation, subject to change, or behind authentication walls. However, such issues might serve as a next logical complication for multimodal LLMs.

*Table 6.* Pass rates (%) by model and programming language.

| Model | Py | JS | Go | Rust | Scala | pass@1 | pass@3 |
|---|---|---|---|---|---|---|---|
| Claude Opus-4.5 | **36.1** | **26.7** | 15.0 | **28.9** | **19.4** | **25.2** | **32.7** |
| GLM 4.7 | 27.2 | 26.1 | **17.2** | 24.4 | 11.7 | 21.3 | 26.7 |
| MiniMax M2.1 | 26.1 | 26.1 | 15.6 | 20.6 | 7.8 | 19.2 | 27.0 |
| Gemini 3 Flash | 25.6 | 25.0 | 11.7 | 21.1 | 7.2 | 18.1 | 27.7 |
| DeepSeek v3.2 | 23.3 | 24.4 | 12.2 | 21.7 | 5.6 | 17.4 | 25.0 |
| GPT 5.2 | 20.6 | 24.4 | 11.1 | 21.7 | 7.2 | 17.0 | 25.0 |
| gpt-oss-120b | 8.9 | 13.3 | 9.4 | 9.4 | 2.8 | 8.8 | 14.3 |

Operating an automated pipeline at scale inevitably introduces environment pathologies that simple static checks cannot catch. Leveraging the findings from this diagnostic study, we implemented the extensive metadata enrichment pipeline described in Section 3.5.

To validate that the diagnostic metadata separates solvable tasks from environment-induced noise, we sampled 60 tasks from Code A and B* categories with similar distributions of language, number of modified files, and number of modified lines. Across all evaluated models, Code A tasks have substantially higher pass rates than B* tasks. For example, Gemini reaches 34.0% pass@3 on Code A but only 4.0% on Code B*, while GLM-4.7 reaches 34.0% versus 6.0%. This gap supports the intended use of the metadata: A-category tasks provide cleaner learning signals, while B-category tasks identify settings where tests, specifications, or environments introduce confounding factors. Full numbers are demonstrated in Table 7.

By automatically tagging instances with diagnostic labels (e.g., B1: TEST_SUITE_COUPLING, B2: IMPLICIT_NAMING, B3: EXTERNAL_DEPENDENCY), we empower researchers to curate training sets based on their specific learning goals:

*Table 7.* Pass @k rates on Code A and B* subsets. Code A denotes tasks classified as cleanly solvable, while Code B* denotes tasks with diagnostic B-category labels.

| Model | A@1 | A@3 | B*@1 | B*@3 |
|---|---|---|---|---|
| DeepSeek V3.2 | 22.0 | 26.0 | 4.0 | 4.0 |
| Gemini | 26.0 | 34.0 | 0.0 | 4.0 |
| GLM 4.7 | 28.0 | 34.0 | 4.0 | 6.0 |
| GPT 5.2 | 14.0 | 26.0 | 4.0 | 6.0 |
| Opus 4.5 | 22.0 | 28.0 | 6.0 | 8.0 |

- **Curriculum Learning:** Filtering out B-category tasks creates a "clean" subset suitable for initial supervised fine-tuning (SFT) or RL warm-up.

- **Robustness Training:** Reintroducing noisy tasks (e.g., B1) at later stages can train agents to handle regression testing and fragile environments, provided the reward function is adjusted (e.g., via partial credit).

- **Context Management:** Tasks tagged with B3 (External Dependencies) can be filtered out for standard training or used specifically to train agents equipped with web-browsing tools.

The full prompt including diagnostic labels is available in Appendix A.3.6. This diagnostic-driven approach ensures SWE-rebench V2 serves as a configurable substrate for training robust software engineering agents.

## 5. Discussion and Limitations

Our goal is to reduce the scarcity of diverse training data for SWE agents, especially for long-tail languages, by enabling scalable collection of executable tasks across many ecosystems. We provide resources that support training and evaluation across a broad set of languages and repositories.

Automated setup and validation at this scale results in some tasks with imperfections, such as the environment preparation issues identified, which can introduce reward noise during training. To mitigate this, we conduct a diagnostic analysis by running multiple models on a subset of tasks. This analysis helps separate failures caused by model capability limitations from those caused by task formulation or environment issues. Based on these observed failure modes, we enrich each task with instance-level diagnostic metadata, enabling users to filter out tasks with known issues or construct subsets tailored to specific research goals.

We outline the following main limitations of our work. First, although SWE-rebench V2 is designed as a training substrate, we do not include end-to-end RL training ablations in this work. A convincing multilingual RL run at the scale enabled by the dataset would require substantial additional infrastructure. Instead, we validate prerequisite properties for training: tasks are executable, non-trivial, show pass@k

headroom across models, and the A/B* analysis shows that metadata separates cleaner tasks from confounded ones. Direct comparisons of filtered versus unfiltered curricula remain important future work.

Second, Docker images make the main dataset robust to repository evolution, but do not eliminate all sources of drift. External package repositories, system packages, and network-hosted resources may change or disappear. We therefore view ongoing maintenance as important and plan periodic rebuilds, environment updates, and metadata revisions.

Third, our current environment design targets projects that can be reproducibly packaged into a single Docker container. This limits coverage of more complex systems where realistic execution requires multiple services, external databases, queues, or distributed infrastructure components.

## 6. Conclusion and Future Work

In this work, we present SWE-rebench V2, a language-agnostic pipeline for constructing executable SWE tasks. The pipeline automates the end-to-end task construction, from mining pull requests to synthesizing executable environments and filtering instances without manual verification. We release 32,079 stable issue-linked tasks from 3,617 repositories across 20 languages, supplemented by 120,000+ PR-derived tasks that expand coverage beyond issue-linked changes. We also provide instance-level diagnostic metadata and ablations on setup synthesis and quality filtering to quantify stage-wise yield and failure modes.

We plan to expand the dataset by increasing setup retries for higher yield, adding curated subsets, and onboarding more long-tail languages. Future work will target extending the pipeline to support complex, long-horizon tasks, such as those in multi-service systems requiring iterative, cross-component modifications. We will also investigate enriching the reward signal beyond test-based correctness to include automatically measurable non-functional requirements like performance, latency, and memory efficiency.

We believe that the automated data collection pipeline and resources released with SWE-rebench V2 provide a practical foundation for training and evaluating LLM-based agents on realistic software engineering tasks at scale.

## Acknowledgements

We thank Nebius Token Factory[1] for providing API credits for model inference.

---

[1]Available at: Nebius Token Factory.

## Impact Statement

This work presents an automated pipeline for constructing large-scale datasets of executable software engineering tasks. By enabling training and evaluation on real-world repositories across multiple programming languages, the released resources may support research on more capable and robust LLM-based software engineering agents.

To mitigate potential risks, all included repositories are sourced from publicly available projects with permissive open-source licenses. We additionally apply automated safety checks, including scans for leaked credentials, to reduce the risk of releasing sensitive information. These measures are intended to ensure responsible data collection and redistribution.

As with any large-scale dataset derived from open-source code, the collected tasks may reflect biases or assumptions present in the underlying repositories. We encourage careful evaluation and responsible use of the released data and pipeline.

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

# A. Supplementary Prompts and Artifacts

## A.1. Repo Filtering Shares by Language

*Table 8.* Share of tasks retained after repository filtering by language.

| Lang. | Tasks | Repos | Kept R. | Kept T. | Task Sh. | Repo Sh. |
|---|---|---|---|---|---|---|
| python | 184,558 | 10,548 | 2,144 | 152,161 | 0.82 | 0.20 |
| go | 123,424 | 6,168 | 1,345 | 104,115 | 0.84 | 0.22 |
| typescript | 114,164 | 7,356 | 1,300 | 92,292 | 0.81 | 0.18 |
| javascript | 80,262 | 7,650 | 915 | 58,349 | 0.73 | 0.12 |
| java | 75,114 | 3,574 | 795 | 64,805 | 0.86 | 0.22 |
| rust | 49,703 | 3,325 | 530 | 39,397 | 0.79 | 0.16 |
| cpp | 41,905 | 2,425 | 478 | 35,099 | 0.84 | 0.20 |
| php | 29,130 | 2,607 | 492 | 23,181 | 0.80 | 0.19 |
| scala | 23,248 | 913 | 913 | 23,248 | 1.00 | 1.00 |
| julia | 14,347 | 922 | 922 | 14,347 | 1.00 | 1.00 |
| c | 13,608 | 1,193 | 1,193 | 13,608 | 1.00 | 1.00 |
| kotlin | 13,003 | 913 | 913 | 13,003 | 1.00 | 1.00 |
| r | 9,368 | 619 | 619 | 9,368 | 1.00 | 1.00 |
| dart | 8,792 | 579 | 579 | 8,792 | 1.00 | 1.00 |
| swift | 7,457 | 809 | 809 | 7,457 | 1.00 | 1.00 |
| elixir | 6,621 | 563 | 563 | 6,621 | 1.00 | 1.00 |
| clojure | 4,815 | 189 | 189 | 4,815 | 1.00 | 1.00 |
| ocaml | 3,089 | 201 | 201 | 3,089 | 1.00 | 1.00 |
| lua | 2,990 | 243 | 243 | 2,990 | 1.00 | 1.00 |

*Table 9.* Project Installation ablations test set

| Language | python | js | js/ts | ruby | go | c | rust | java | php | cpp |
|---|---|---|---|---|---|---|---|---|---|---|
| **Tasks/Repos** | 55 | 11 | 6 | 6 | 5 | 5 | 5 | 4 | 4 | 2 |

## A.2. Base Dockerfile Example

*Listing 1.* Base Dockerfile example used for Julia tasks

```
FROM --platform=linux/amd64 julia:1.10-bookworm

ARG DEBIAN_FRONTEND=noninteractive
ENV TZ=Etc/UTC

# Base tools + typical build deps many Julia packages use
RUN apt-get update && apt-get install -y --no-install-recommends \
    git \
    curl \
    wget \
    unzip \
    zip \
    ca-certificates \
    build-essential \
    pkg-config \
    gfortran \
    cmake \
    gnupg \
    libssl-dev \
    && rm -rf /var/lib/apt/lists/*

# Useful Julia env defaults
ENV JULIA_NUM_THREADS=auto \
    JULIA_PKG_SERVER=https://pkg.julialang.org \
    MPIR_CVAR_CH3_INTERFACE_HOSTNAME=127.0.0.1 \
    JULIA_MPIEXEC_FLAGS="-hosts localhost"
```

```
RUN adduser --disabled-password --gecos 'dog' nonroot

# Working directory for your Julia projects
WORKDIR /workspace

# Keep Julia's depot (package cache) inside workspace so it's writable even for non-root
ENV JULIA_DEPOT_PATH=/workspace/.julia
```

## A.3. Prompt Templates and Examples

### A.3.1. BASE DOCKERFILE GENERATOR PROMPT

```
You are an expert DevOps/build engineer. Your task is to generate a minimal,
    reusable base Dockerfile for projects written in <LANG_NAME>.

Goal
Create a base image that contains the runtime/toolchain and common system
    dependencies needed to:
1) install project dependencies,
2) build/compile the project (if applicable),
3) run the project's test suite.

Scope and constraints
- This is a *base* image used across many repositories, so avoid repository-
    specific assumptions.
- Prefer official or widely used upstream base images for <LANG_NAME> (e.g.,
    official images on Docker Hub) rather than installing the toolchain from
    scratch.
- Target linux/amd64 unless <LANG_NAME> strongly requires otherwise.
- Use only system packages that are broadly useful (git, curl, ca-certificates,
    build tools, common native libs). Avoid adding uncommon services.
- Do not add project source code, do not run project commands, and do not require
    network access at runtime beyond typical dependency installation.

Required contents
1) A suitable FROM line (prefer a stable, commonly used toolchain version; choose
    an LTS/stable version when possible).
2) Installation of common build and debugging utilities:
    - git, curl/wget, unzip/zip, ca-certificates
    - build-essential / compiler toolchain, pkg-config, cmake (if relevant)
3) Installation of native libraries commonly required by packages in this
    ecosystem (keep this conservative).
4) Environment variables that place language/package-manager caches under /
    workspace (so they are writable even for non-root):
    - Set XDG_CACHE_HOME, XDG_CONFIG_HOME, XDG_DATA_HOME when applicable
    - Also set language-specific cache/home vars (e.g., for package managers)
5) Create a non-root user (but keep USER optional, as a commented line).
6) WORKDIR /workspace.

Output requirements
- Return only the Dockerfile contents, and nothing else (no explanation, no
    markdown).
- Keep the Dockerfile reasonably short and readable.
- Clean apt caches (rm -rf /var/lib/apt/lists/*) to reduce image size.

Example style (Elixir)
[The following is only an example of the expected style and structure; do not copy
    it verbatim.]
```

```
FROM --platform=linux/amd64 elixir:1.16
ARG DEBIAN_FRONTEND=noninteractive
ENV TZ=Etc/UTC
RUN apt-get update && apt-get install -y --no-install-recommends \
    git curl wget unzip zip ca-certificates \
    build-essential pkg-config cmake gnupg \
    libssl-dev zlib1g-dev libgmp-dev libffi-dev \
 && rm -rf /var/lib/apt/lists/*
ENV MIX_HOME=/workspace/.mix \
    HEX_HOME=/workspace/.hex \
    XDG_CACHE_HOME=/workspace/.cache \
    XDG_CONFIG_HOME=/workspace/.config \
    XDG_DATA_HOME=/workspace/.local/share
RUN adduser --disabled-password --gecos 'nonroot' nonroot
# USER nonroot
WORKDIR /workspace

Now generate the base Dockerfile for {{lang}}
```

## A.3.2. PROMPT FOR SETUP SYNTHESIS

```
# ROLE
You are a non-interactive build-and-test agent with terminal access. The repo is
    already checked out in the current working directory.

# OBJECTIVE
1) Discover how to install/build the project and run its tests.
2) Execute the minimal steps to make tests runnable.
3) Generate a correct `install_config.json` with exactly:
- "install": ordered list of the **actual** successful shell commands required for
     this repo.
- "test_cmd": a list of shell commands to run the full test suite (**per-test
    verbose**, no ANSI color if possible) that is **general** (no file-, directory
    -, or target-specific selectors). Should also output XML if it is possible.
- Add quiet/silent flags to all install/package-manager commands (when supported)
    to minimize logs while still surfacing errors (e.g., --quiet, -q, --silent,
    disable progress bars). Example: `apt-get install -y -qq build-essential`.

# CAPABILITIES & CONSTRAINTS (RULES)
- Be fully non-interactive: `export DEBIAN_FRONTEND=noninteractive`; use `-y`
    flags; **no** `sudo`.
- Outputs of the commands will be truncated to 64 lines of 500 characters each.
- Prefer documented commands from README/Makefile/CI over heuristics.
- **`test_cmd` MUST be per-test verbose (show individual test names).** Always add
     the runner's per-test verbosity flags.
- If the project uses `make test` or other build systems to run tests, extract the
     actual command from the Makefile and use it as `test_cmd`, **with verbose
    flags** if possible.
- **Locate the project root**: if the repository is a monorepo, `cd` into the
    subdirectory that contains the primary build file (e.g., `package.json` / `
    pyproject.toml` / `go.mod` / `Cargo.toml` / `pom.xml`) **before** installing
    or testing. Include that `cd` in `"install"` if required.
- **Use lockfile-aware installs** where available (e.g., `npm ci`, `pip install -r
     requirements.txt` / `uv pip sync`, `go mod download`, `cargo fetch`) and
    prefer vendored dependencies if present. If a private registry/module or
    credentials are required, **stop** with a short note rather than hanging or
```

           adding auth.
- Only include commands in "install" that were **necessary** and **succeeded** in
    this run (no stray `cd`, `rm`, or env exports unless required). **Do not** add
     commands that failed or were later replaced.
- **Do not** put language/runtime *installation* commands into `install_config.
    json`; keep it repo-scoped.
- **Set minimal env vars** required by the repo's docs/CI for tests (safe dummy
    values if needed) and record those `export` lines in `"install"`.
- **Verify test discovery** from the chosen root (e.g., a short list/collect mode)
     before running the full suite; then run the full suite with verbosity.
- If `.gitmodules` exists: `git submodule update --init --recursive` (only if
    needed).
- Keep output short: prefer quiet flags; use `--no-color`/`--color=never` where
    available.
- If test results are saved to a file (e.g., XML, HTML, or other formats), add a
    command to output the test results to stdout (such as `cat <result_file>`) and
     append it to `test_cmd` so that test results are visible in the output.
- If tests run but fail, you may still record the working **verbose** `test_cmd`.
- For noisy commands (especially building the project), prefer: ... 2>&1 | tail -n
     30 (or a tool's --tail option, if it exists) to show only the last lines
    while preserving the exit code.
- If the test runner or framework caches results (e.g., pytest, tox, or similar),
    ensure that the `test_cmd` includes a command to clear or ignore the cache
    before running tests, so that tests are always executed fresh. For example,
    for pytest, use `-p no:cacheprovider` to disable the cache plugin, or for
    other frameworks, add the appropriate cache-cleaning command before running
    the tests.
- You should ensure that test logs contain individual test names (exact names) and
     their status (e.g., pass/fail/skip) and don't contain ANSI color codes. After
     running the test command, verify the output includes each test's exact name
    and status. Try to add verbosity and no-color flags to the test command if
    needed.
- You do not need coverage options for `test_cmd`.
- If test results are saved to XML files (e.g., Gradlew, JUnit, TestNG), add a
    command to find and print all XML reports to stdout after running tests, e.g.:
     `find . -name '*.xml' -exec cat {} + || true` and append it to `test_cmd`
    using `&&` or `||` as appropriate, so test results are always visible in the
    output.
- If you need to modify repository files, do so only with non-interactive sed
    commands (example: sed -i 's/OLD/NEW/' relative/path). Record each successful
    sed command exactly as run in the "install" array (these edits count as
    necessary install steps).
- Do NOT create install_config if you can't build the project!

# SPECIAL COMMANDS
You may also use the following special commands when applicable:
{{ tools_info | map('replace', '\n\n', '\n') | join('\n') }}

# RESPONSE STYLE
For each action, reply with a **one-line plan** followed by **one shell command**
    to execute.
When finished, create `install_config.json` via a heredoc and print it with `cat`.
Your **final output must be exactly one fenced JSON code block** containing only
    the three fields above. No extra text after it.

# ENVIRONMENT PROMPT
The shell prompt shows current directory and file like:
```
(Current directory: <dir>, current file: <file>) bash-$
```

So that you always know what the current directory is and what file is currently

```
    open.

# IMPORTANT
- Always use VERBOSE test runners to ensure individual test names are shown in the
    output. Check it after running tests. And update `test_cmd` if needed.

# GOAL
Install the project and run its tests.

You are installing the repo: {{issue_description}}

# HOW TO WORK (CHECKLIST)
1) Discover: read top-level docs (`README*`, `docs/*`, `CONTRIBUTING*`),
    automation files (`Makefile`, `package.json`, `pyproject.toml`, `go.mod`, `
    Cargo.toml`, `pom.xml`, `build.gradle*`, `tox.ini`, etc.), and CI configs (`.
    github/workflows/*`, `.travis.yml`, etc.)-but stop as soon as you find enough
    information to install the project or run the tests.
2) Plan minimal install: **identify the project root first** (subdir if monorepo);
     then project deps/build via the repo's package manager (**lockfile-aware
    commands** preferred).
3) Execute iteratively: after each **successful and necessary** command, add it (
    in order) to `"install"`. **Do not** include commands that failed or were not
    needed.
4) Choose a single, **per-test verbose** and **no-color** test runner for `"
    test_cmd"`, and it must be **general** (no file-/target-specific selection).
    If `make test` is presented **extract the underlying test command and add per-
    test verbosity**. Should use JunitXML output if possible.
5) Produce the install_config.json file if you have a working `test_cmd` (even if
    some tests fail):

```bash
cat > install_config.json <<'JSON'
{
    "install": [ ... ],
    "test_cmd": [ ... ]
}
JSON
cat install_config.json
```

You should create install_config.json file only if you successfully ran tests or
    if you have a working `test_cmd` that runs the full suite (even if some tests
    fail).
If you can't build the project or run tests, don't create install_config and just
    submit.

Repository is uploaded; your shell is at the repo root.
```

### A.3.3. LOG PARSER GENERATOR PROMPT

```
You are given raw stdout/stderr from a repository test run. Your task is to write
    a Python log parser that converts this log into a structured per-test status
    map.

Output format
- Implement a single function:
```

```
        def parse_log(log: str) -> dict[str, str]:

- The function must return a dictionary mapping each test case name to one of:
    TestStatus.PASSED.value
    TestStatus.FAILED.value
    TestStatus.SKIPPED.value

Assumptions
- You will be provided (1) one or more example logs produced by the same test
    runner, and (2) a short description of how tests are invoked (optional).
- Test names should be stable identifiers (e.g., suite/class + method, or fully-
    qualified test name) as they appear in the log.
- If the log contains repeated entries for the same test, the final observed
    outcome should be used.

Requirements
- Use only Python standard library modules (e.g., re, typing). Do not import
    external dependencies.
- The parser must be robust to noise lines, timestamps, and unrelated output.
- Prefer parsing from structured markers when available (e.g., explicit PASS/FAIL/
    SKIP lines), rather than relying on summary lines.
- If the runner prints parameterized tests or subtests, include the full name used
     by the runner (including parameters) as the key.

Implementation guidance
- Use regular expressions or simple string matching to extract (status, test_name).

- Ignore infrastructure failures that do not correspond to a specific test case
    unless the runner clearly reports them as a named test.

Example (Go test)
Below is an example of the expected style and return type for 'go test'-like
    output:

def parse_log_gotest(log: str) -> dict[str, str]:
    test_status_map = {}
    pattern = r"^--- (PASS|FAIL|SKIP): (.+) \((.+)\)$"
    for line in log.split("\n"):
        m = re.match(pattern, line.strip())
        if m:
            status, test_name, _ = m.groups()
            if status == "PASS":
                test_status_map[test_name] = TestStatus.PASSED.value
            elif status == "FAIL":
                test_status_map[test_name] = TestStatus.FAILED.value
            elif status == "SKIP":
                test_status_map[test_name] = TestStatus.SKIPPED.value
    return test_status_map

Now write the parser for the provided test log(s). Return only Python code
    containing the parse_log function (and any helper definitions it needs), and
    nothing else.
```

## A.3.4. PROBLEM STATEMENT GENERATOR PROMPT

```
You are a senior engineer writing a pull request description.

## Input
You will receive a code patch (diff) and the original description. Use it only to
    infer the problem and required behavior. Do NOT describe the implementation.

## Goal
Produce a PR description that explains the problem and what correctness requires,
    without revealing details unique to this patch.

## Rules
- Do NOT mention file names, symbols, line numbers, commit hashes, or code
    snippets.
- Do NOT describe algorithms, control flow, or data structures.
- Avoid code-level "changed X to Y" statements.
- High-level component references are allowed.
- If intent is unclear, state a brief assumption.

## Format
- **Title** -- one line.
- **Problem** -- 2-3 sentences max.
- **Root Cause** -- 1-2 sentences, conceptual.
- **Fix / Expected Behavior** -- bullet list, max 5 bullets.
- **Risk & Validation** -- 2-3 bullets total.

## Style Constraints
- Prefer bullet points over paragraphs.
- No section should exceed 3 lines.
- Omit anything non-essential to understanding or reviewing the change.

## Output
Return only the PR description.

---

## Inputs

**REPO**: {{ repo }}

**ORIGINAL DESCRIPTION**
```
{{ pr_description }}
```

**PATCH**:
```
{{ patch }}
```
```

## A.3.5. LOG PARSER EXAMPLE

*Listing 2.* Log parser example for ExUnit output

```
def parse_log_elixir(log: str) -> dict[str, str]:
    """Parse ExUnit output and return {full_test_name: status}.

    Rules:
      * Lines like: "* test <name> [L#42]" or with timing "(12.3ms)" -> PASSED (tentative)
```

```
    * Lines like: "* test <name> (skipped) [L#42]" -> SKIPPED
    * Failure headers: "1) test <name> (<Module>)" -> FAILED (overrides prior PASS)
"""
results: dict[str, str] = {}

# Regexes
skipped_re = re.compile(r"^\\*\\s+test\\s+(.*?)\\s+\\(skipped\\)\\s+\\[L#\\d+\\]$")
passed_timed_re = re.compile(
    r"^\\*\\s+test\\s+(.*?)\\s+\\([0-9]+(?:\\.[0-9]+)?ms\\)\\s+\\[L#\\d+\\]$"
)
passed_basic_re = re.compile(r"^\\*\\s+test\\s+(.*?)\\s+\\[L#\\d+\\]$")
failure_header_re = re.compile(r"^\\d+\\)\\s+test\\s+(.*?)\\s+\\([^)]+\\)$")

for raw in log.splitlines():
    line = raw.strip()
    if not line:
        continue
    if m := skipped_re.match(line):
        results[m.group(1)] = TestStatus.SKIPPED.value
        continue
    if m := failure_header_re.match(line):
        results[m.group(1)] = TestStatus.FAILED.value
        continue
    if m := passed_timed_re.match(line):
        results.setdefault(m.group(1), TestStatus.PASSED.value)
        continue
    if m := passed_basic_re.match(line):
        results.setdefault(m.group(1), TestStatus.PASSED.value)
        continue
return results
```

## A.3.6. PROMPT FOR METADATA ENRICHMENT

```
You are classifying software engineering tasks for RL validity. Determine if a
    task is of high-quality or reflects environment preparation issues (B1-B7).

## Classification Codes

| Code | Category | Description |
|------|----------|-------------|
| A | SOLVABLE | Problem is clearly specified, tests align with stated
    requirements |
| B1 | TEST_SUITE_COUPLING | Correct fix may fail due to unrelated test
    regressions |
| B2 | IMPLICIT_NAMING | Tests expect specific names/signatures not in problem
    statement |
| B3 | EXTERNAL_DEPENDENCY | Essential info in external URLs not quoted in issue |
| B4 | AMBIGUOUS_SPEC | Missing expected behavior, repro steps, or acceptance
    criteria |
| B5 | PATCH_ARTIFACTS | Reference patch includes unrelated changes tests may
    require |
| B6 | IMPLICIT_KNOWLEDGE | Requires domain knowledge not stated or inferrable
    from repo |
| B7 | INLINE_TEST | Inline test in the non-test patch |

## Decision Process

1. Extract intent from ISSUE_TEXT: expected behavior, observed behavior,
    reproduction, acceptance criteria
2. Check test alignment: Do assertions verify stated requirements or introduce new
```

```
        ones?
3. Scan for B-category signals:
   - B1: Tests span unrelated modules
   - B2: String assertions for names not in issue
   - B3: URLs to external docs/specs
   - B4: Vague or missing acceptance criteria
   - B5: Patch modifies unrelated files that are then required by tests
   - B6: Tests assert specific approaches not mentioned
   - B7: Inline tests appear in the non-test patch
4. Classify: A if clean, B[1-7] for primary issue
```

## Rubric: PR Category

Choose one or more categories that best describe the PR.

Allowed categories:

```
{
  "critical_bug", "major_bug", "minor_bug", "regression_bug", "edge_case_bug", "
    performance_bug",
  "security_bug", "integration_feat", "core_feat", "ui_ux_feat", "dev_ops_enh", "
    documentation_enh"
}
```

## Rubric: Task Difficulty

Estimate task difficulty based on the amount of code changes required, logic
    complexity, and domain knowledge needed. Choose one level assuming expected
    time to implement: easy (<15 min), medium (15 min - 1h), hard (>1h).

Allowed difficulty levels:

```
{
  "easy", "medium", "hard"
}
```

## Output (JSON only)

```json
{
  "reasoning": "3-5 sentences: (1) summarize issue intent, (2) assess test
    alignment, (3) note any B-category signals, (4) justify classification",
  "intent_completeness": "complete | partial | insufficient",
  "test_alignment_issues": ["list specific misalignments or empty if aligned"],
  "detected_issues": {
    "B1": false, "B2": false, "B3": false, "B4": false,
    "B5": false, "B6": false, "B7": false
  },
  "external_urls": ["extracted URLs if any"],
  "pr_categories": ["one or more from the PR category rubric"],
  "difficulty": "easy | medium | hard",
  "code": "A|B1|B2|B3|B4|B5|B6|B7",
  "confidence": 0.0
}
```

```
---

## Inputs

**REPO**: {{ repo }}

**ISSUE_TEXT**:
```
{{ problem_statement }}
```

**TEST_PATCH**:
```
{{ test_patch }}
```

**GOLDEN PATCH** (reference only):
```
{{ patch }}
```
```

### A.3.7. PROMPT FOR INTERFACE GENERATION

```
You analyze code changes in pull requests. Your task is to extract information
    about changed interfaces and signatures that are explicitly checked in the
    tests.

## Decision Process

1. Identify new functions or classes
2. Identify functions or classes whose signatures have changed significantly (
    arguments changed, not just indentation)
3. Use the test patch to see which of them, if any, are explicitly called in the
    new test code

Ignore as interfaces:

- Purely internal helpers (for example names starting with "_" and not used
    directly in tests)
- Type aliases, enums, simple data or config holders
- Constructors or trivial overrides that only support other documented interfaces
- Constants, flags, small config values, and simple data declarations
- Anything that is not explicitly called in the test patch

## Output Format

Output must be wrapped in a single root tag:

```
<ANSWER>Method: <ClassName.method_name(self, key_params...)> Location:  Inputs: <
    important parameters, types or roles, and key constraints> Outputs: <return
    type or behavior, including important error conditions> Description: <1 to 2
    sentences on what it does and when to use it>

Function: <function_name(key_params...)>
Location:
Inputs: <important parameters, types or roles, and key constraints>
```

```
Outputs: <return type or behavior, including important error conditions>
Description: <1 to 2 sentences on what it does and when to use it>

...repeat blocks for each interface...</ANSWER>
```

If there are no qualifying interfaces, output exactly:

```
<ANSWER>No new interfaces are introduced.</ANSWER>
```

### A.3.8. PROMPT FOR FILTERING BY ISSUE CLARITY

```
We have a dataset of GitHub issues from various open-source Python repositories.
    Each issue comes with a PR that successfully solves the issue described. Each
    PR consists of 2 parts: (1) code that resolves the issue (2) changes to the
    test files of the repository, which check whether the issue has been resolved.

We intend to use samples in this dataset as a benchmark for coding ability: For
    each sample, we give an engineer the issue text and ask them to write code to
    resolve the issue (without revealing the solution from the original PR). Then,
     we apply the test files from the original PR to their code and run the tests
    to check whether their solution passes.

Importantly, this setup assumes that:
- The issue description is sufficiently well-specified to understand what the
    problem is, and what the correct solution should look like.
- The tests are correctly scoped to the issue i.e. they correctly test for a
    solution exactly as described in the issue description, and do not test any
    other unrelated functionality.

In this task, you will help to check those assumptions and identify which issue +
    test samples are suitable for use in our benchmark.

You are now considering an issue from the {{ repo }} repository.

Please take a moment to read the issue description below.

<problem_statement>
{{ problem_statement }}
</problem_statement>

In your assessment, you may refer to the contents of the original PR given below.

<patch>
{{ patch }}
</patch>

<test_patch>
{{ test_patch }}
</test_patch>

How well-specified is the issue text? Imagine that you are an experienced software
     engineer who has been instructed to create a PR that successfully resolves
    the above GitHub issue. You have full access to the codebase, and can see the
    issue description as it is above. But you are not able to ask for
```

```
   clarification and would need to work exclusively from this information. Is the
      issue description well-specified enough for a meaningful attempt at a
      solution?

You should classify this issue based on the following options.

- 0: The issue is well-specified and it is clear what is required for a successful
      solution.
<example>
{{ examples[0].problem_statement -}}
</example>

- 1: There are some blanks to fill in about the issue, but there is a sensible
      interpretation of what is required for a successful solution.
<example>
{{ examples[1].problem_statement -}}
</example>

- 2: The issue is vague and there is room for ambiguity. It is unclear what a
      successful solution would look like.
<example>
{{ examples[2].problem_statement -}}
</example>

- 3: It is almost impossible to understand what you are being asked to do without
      further information.
<example>
{{ examples[3].problem_statement -}}
</example>

Please respond with a short explanation of your choice (minimum 100 characters)
      followed by the selected option number.
```

## B. Additional Plots

This section provides supplementary distributional views of the benchmark corpus. The year and language histograms contextualize when issues were reported and which ecosystems dominate the dataset, complementing the main paper's dataset summary.

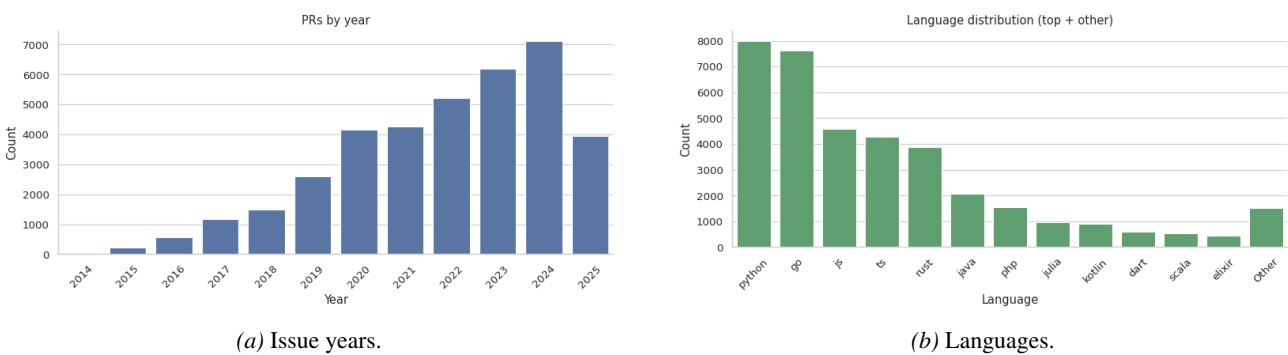

*(a)* Issue years.                    *(b)* Languages.

*Figure 1.* Temporal and language distributions in the benchmark corpus.

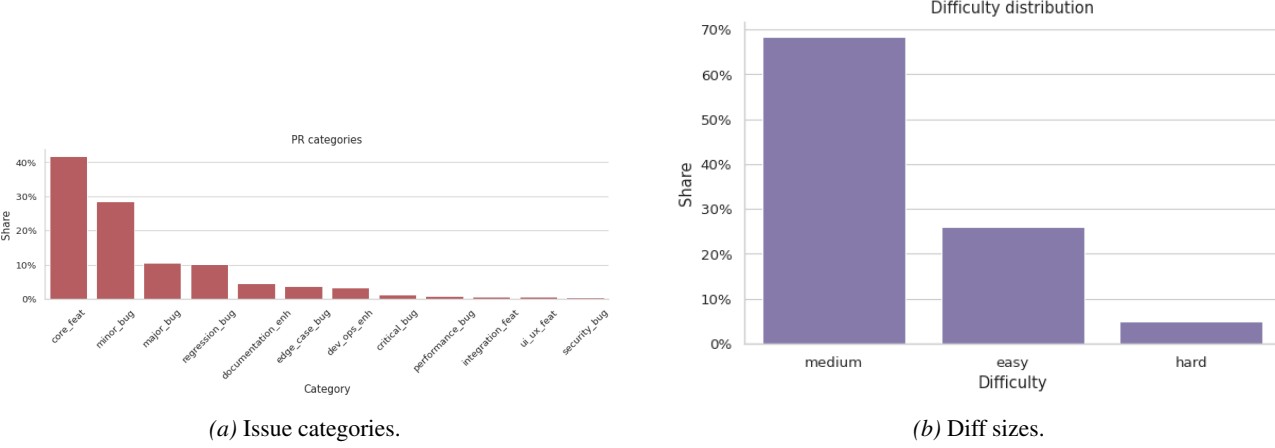

*(a)* Issue categories.

*(b)* Diff sizes.

*Figure 2.* Issue-type mix and patch-size distribution.

## C. Model Performance Results Across Languages

### C.1. Per-language Performance with Confidence Intervals

*(a)* Go: pass@1, SEM, 95% CI, and pass@3 (60 tasks).

| Model | pass@1 | SEM | CI95% | pass@3 |
|---|---|---|---|---|
| GLM-4.7 | **17.22%** | 4.38% | [8.65%, 25.80%] | 23.33% |
| MiniMax-M2.1 | 15.56% | 4.15% | [7.42%, 23.69%] | 23.33% |
| Opus-4.5 | 15.00% | 3.92% | [7.33%, 22.67%] | **25.00%** |
| DeepSeek-V3.2 | 12.22% | 3.88% | [4.62%, 19.82%] | 16.67% |
| Gemini | 11.67% | 3.25% | [5.30%, 18.04%] | 21.67% |
| GPT-5.2 | 11.11% | 3.23% | [4.77%, 17.45%] | 20.00% |
| gpt-oss-120b | 9.44% | 3.55% | [2.48%, 16.41%] | 11.67% |

*(b)* JavaScript: pass@1, SEM, 95% CI, and pass@3 (60 tasks).

| Model | pass@1 | SEM | CI95% | pass@3 |
|---|---|---|---|---|
| Opus-4.5 | **26.67%** | 5.42% | [16.04%, 37.29%] | 31.67% |
| MiniMax-M2.1 | 26.11% | 5.14% | [16.04%, 36.19%] | **35.00%** |
| GLM-4.7 | 26.11% | 5.44% | [15.45%, 36.77%] | 30.00% |
| Gemini | 25.00% | 4.99% | [15.22%, 34.78%] | 33.33% |
| DeepSeek-V3.2 | 24.44% | 5.13% | [14.40%, 34.49%] | 31.67% |
| GPT-5.2 | 24.44% | 5.00% | [14.64%, 34.25%] | 33.33% |
| gpt-oss-120b | 13.33% | 3.65% | [6.18%, 20.48%] | 21.67% |

*(a)* Python: pass@1, SEM, 95% CI, and pass@3 (60 tasks).

| Model | pass@1 | SEM | CI95% | pass@3 |
|---|---|---|---|---|
| Opus-4.5 | **36.11%** | 6.20% | [23.95%, 48.27%] | **36.67%** |
| GLM-4.7 | 27.22% | 5.46% | [16.52%, 37.92%] | 31.67% |
| MiniMax-M2.1 | 26.11% | 5.32% | [15.68%, 36.54%] | 31.67% |
| Gemini | 25.56% | 5.16% | [15.45%, 35.66%] | 33.33% |
| DeepSeek-V3.2 | 23.33% | 4.97% | [13.60%, 33.07%] | 31.67% |
| GPT-5.2 | 20.56% | 4.95% | [10.85%, 30.26%] | 23.33% |
| gpt-oss-120b | 8.89% | 2.84% | [3.32%, 14.46%] | 16.67% |

*(b)* Rust: pass@1, SEM, 95% CI, and pass@3 (60 tasks).

| Model | pass@1 | SEM | CI95% | pass@3 |
|---|---|---|---|---|
| Opus-4.5 | **28.89%** | 5.34% | [18.42%, 39.36%] | **38.33%** |
| GLM-4.7 | 24.44% | 5.25% | [14.16%, 34.73%] | 30.00% |
| GPT-5.2 | 21.67% | 4.61% | [12.64%, 30.70%] | 33.33% |
| DeepSeek-V3.2 | 21.67% | 4.47% | [12.91%, 30.43%] | 33.33% |
| Gemini | 21.11% | 4.41% | [12.47%, 29.75%] | 33.33% |
| MiniMax-M2.1 | 20.56% | 4.56% | [11.62%, 29.49%] | 30.00% |
| gpt-oss-120b | 9.44% | 3.08% | [3.41%, 15.48%] | 16.67% |

*(a)* Scala: pass@1, SEM, 95% CI, and pass@3 (60 tasks).

| Model | pass@1 | SEM | CI95% | pass@3 |
|---|---|---|---|---|
| Opus-4.5 | **19.44%** | 4.29% | [11.04%, 27.85%] | **31.67%** |
| GLM-4.7 | 11.67% | 3.62% | [4.58%, 18.75%] | 18.33% |
| MiniMax-M2.1 | 7.78% | 2.55% | [2.78%, 12.78%] | 15.00% |
| GPT-5.2 | 7.22% | 2.52% | [2.29%, 12.16%] | 15.00% |
| Gemini | 7.22% | 2.25% | [2.80%, 11.64%] | 16.67% |
| DeepSeek-V3.2 | 5.56% | 2.26% | [1.12%, 9.99%] | 11.67% |
| gpt-oss-120b | 2.78% | 1.64% | [-0.44%, 5.99%] | 5.00% |

*(b)* All: pass@1, SEM, 95% CI, and pass@3 (300 tasks).

| Model | pass@1 | SEM | CI95% | pass@3 |
|---|---|---|---|---|
| Opus-4.5 | **25.00%** | 2.00% | [21.00%, 30.00%] | **33.00%** |
| GLM-4.7 | 21.00% | 2.00% | [17.00%, 26.00%] | 27.00% |
| MiniMax-M2.1 | 19.00% | 2.00% | [15.00%, 23.00%] | 27.00% |
| Gemini | 18.00% | 2.00% | [14.00%, 22.00%] | 28.00% |
| DeepSeek-V3.2 | 17.00% | 2.00% | [14.00%, 21.00%] | 25.00% |
| GPT-5.2 | 17.00% | 2.00% | [13.00%, 21.00%] | 25.00% |
| gpt-oss-120b | 9.00% | 1.00% | [6.00%, 11.00%] | 14.00% |

