# OpenReview forum: "SWE-rebench V2: Language-Agnostic SWE Task Collection at Scale"
_ICML.cc/2026/Conference — ICML 2026 regular_

### Official Review · Reviewer_o4vH · 2026-02-26

**Soundness:** 3
**Presentation:** 3
**Significance:** 2
**Originality:** 2
**Overall Recommendation:** 4
**Confidence:** 4

**Summary:**

SWE-Rebench v2 is an automatic pipeline for constructing large-scale executable software engineering tasks. It contains 36k tasks across 20 languages and 3.8k repos, with 100k total tasks using generated problem statements. To my knowledge, this is the largest and most diverse set of environments, and serves as a valuable training or evaluation set.

**Compliance With Llm Reviewing Policy:**

Affirmed.

**Final Justification:**

see rebuttal

**Key Questions For Authors:**

Since the main motivation and strength is the use of this dataset as a training set, can additional experiments be done in this area? For example, which environments (out of the 20k) are most effective for RL? Does training on this dataset improve model performance or generalization? I think it would be quite beneficial to see these experiments.

**Limitations:**

Yes

**Strengths And Weaknesses:**

Strength:
- The size of the benchmark. It covers more than 20 languages across many repos, making it significantly more diverse compared to SWE-Bench Verified or SWE-Bench Pro.
- There is a significant engineering contribution in running the task collection, which will be valuable for the community.
Weaknesses:
- I fear there is a lack of novelty. Many task collection pipelines exist (e.g. SWE-Smith, Multi-SWE-Bench) and it's not clear what the novelty here is compared to those pipelines.
- The dataset will suffer from contamination issues. While it can still be valuable as an open-source training dataset, there were not experiments in this area.
- There are some weaknesses due to a lack of verification, which the authors acknowledge. (e.g. tests can be too broad or too specific)

---

> ### Author Rebuttal · Authors · 2026-03-31
>
> We thank the reviewer for the thoughtful feedback. We address the main concerns below.
>
> While we acknowledge the excellent recent progress in automated task collection (such as SWE-Smith and Multi-SWE-Bench), our contribution is not a single new pipeline component in isolation, but rather the combination of several elements into a unified, training-oriented resource at substantially larger scale. In particular, we target broad language coverage under a single executable contract, release training-oriented open-source artifacts for reproducible interactive learning, including pre-built environments, and provide instance-level and trajectory-level diagnostics embedded into the metadata of each task. We also release the full pipeline code. We believe this combination is the key novelty of the work. As stated in the paper, “SWE-rebench V2 builds on these efforts by targeting language breadth under a single executable contract, by releasing training-oriented artifacts for reproducible interactive learning, including pre-built environments, and by rigorous instance-level diagnostics.”
>
> Regarding contamination, we agree that this is an important concern for evaluation benchmarks. At the same time, our primary intended use case is training rather than leaderboard-style evaluation, and in that setting contamination is less problematic than for a held-out benchmark intended to measure clean generalization. Our goal is for this dataset to support large-scale post-training of software engineering agents, including RL-style training.
>
> We also agree that lack of perfect verification remains a limitation. As the reviewer notes, tests can sometimes be too broad or too narrow, and automatically constructed environments can contain noise. We attempted to address these issues directly in Sections 3 and 4 by introducing a failure mode taxonomy, providing metadata for quality-related dimensions, and including task interfaces and diagnostic signals that help users identify and filter problematic instances. Rather than assuming that all automatically generated tasks are equally reliable, we make these sources of imperfection explicit and actionable.
>
> Finally, we agree that additional training-oriented experiments would strengthen the paper substantially. As we noted in our responses to other reviewers, the most direct validation would be a downstream training experiment showing which environments are most effective for RL and whether training on this dataset improves model performance or generalization. We believe such experiments are highly valuable and align directly with our long-term goal for this resource. At the same time, conducting them at meaningful scale would require substantial additional compute and infrastructure. During the rebuttal period, however, we prepared an additional ablation comparing solved rates on A-category versus B-category tasks (randomly sampled 60 tasks). This analysis helps show that A-category tasks have higher solved rates, consistent with reduced environmental noise and better suitability for training. We will include these results in the final manuscript. The results are as follows:
>
> Code A:
> | Model        | Passed 1 | Passed 2 | Passed 3 |
> |--------------|---------:|---------:|---------:|
> | deepseek_v32 | 22.0%    | 26.0%    | 26.0%    |
> | gemini       | 26.0%    | 32.0%    | 34.0%    |
> | glm_47       | 28.0%    | 30.0%    | 34.0%    |
> | gpt_52       | 14.0%    | 24.0%    | 26.0%    |
> | opus_45      | 22.0%    | 28.0%    | 28.0%    |
>
> Code B*
> | Model        | Passed 1 | Passed 2 | Passed 3 |
> |--------------|---------:|---------:|---------:|
> | deepseek_v32 | 4.0%     | 4.0%     | 4.0%     |
> | gemini       | 0.0%     | 2.0%     | 4.0%     |
> | glm_47       | 4.0%     | 4.0%     | 6.0%     |
> | gpt_52       | 4.0%     | 4.0%     | 6.0%     |
> | opus_45      | 6.0%     | 6.0%     | 8.0%     |

---

> > ### Author Rebuttal · Reviewer_o4vH · 2026-04-02
> >
> > Thanks to the authors for the additional data and information, and the detailed response. I will raise my score, since I agree that it is difficult to run the RL at scale and the dataset is quite a value contribution. However, it could be possible to run a large-scale pass@16 run which can be used in lieu of an RL rollout. Even this for a smaller model (eg one trained once SWE tasks, so small model but good performance) I think would help to enhance the impact of the work.

---

### Official Review · Reviewer_EoAC · 2026-03-12

**Soundness:** 2
**Presentation:** 3
**Significance:** 4
**Originality:** 2
**Overall Recommendation:** 5
**Confidence:** 3

**Summary:**

This paper presents SWE-rebench V2, an automated pipeline for constructing large-scale, executable software engineering tasks for reinforcement learning training. The authors address the concept of language-agnostic environment construction, releasing 36,000+ containerized tasks spanning 20 languages and 3,800+ repositories, plus 100,000+ PR-derived tasks with synthetic problem statements. The pipeline combines an interactive setup agent, ensemble LLM-based quality filtering, and instance-level diagnostic metadata.

**Compliance With Llm Reviewing Policy:**

Affirmed.

**Final Justification:**

The authors have chosen to leave most of my suggestions for future work, which is entirely understandable. I would like to emphasize the significance of this work, as I believe the proposed dataset is extremely valuable for practitioners. I maintain my current score.

**Key Questions For Authors:**

See weaknesses

**Limitations:**

yes

**Strengths And Weaknesses:**

# Strengths

1. The dataset significantly extends prior multilingual SWE corpora spanning 20 languages and 3,800+ repositories, this is a notable contribution to the SWE agents.
2. The release includes pre-built Docker images, fail-to-pass test oracles, and rich diagnostic metadata. This artifacts are practically necessary for reproducible RL training but frequently omitted in prior work.
3. The comparisons of interactive vs. non-interactive setup agents, ensemble filtering strategies and model choices are well-constructed and provide actionable guidance for pipeline design choices.
4. The 100,000+ PR-derived tasks address issue linkage, with leakage mitigation measures described.

# Weaknesses

1. The paper claims to release datasets and pre-built artifacts, but supplementary materials contain only code.
2. Missing downstream training validation. This is the paper's most significant gap. The diagnostic metadata and curriculum design proposals are well-motivated but entirely unevaluated. No ablation is presented showing that filtering by B-category tags, using generated interfaces, or applying curriculum ordering actually improves agent training outcomes. The authors acknowledge this in Section 5 but frame it as "out of scope" — for a paper whose primary contribution is a training dataset, this is a substantial omission. A reviewer expects at least one training run comparing filtered vs. unfiltered data subsets.
3. Setup synthesis success rate is underanalyzed. Table 1 reveals that only ~20% of repositories succeed with a single setup attempt. While the authors note this and argue it reflects the difficulty of the task, there is no breakdown of failure modes by language or repository type. Understanding why setup fails — wrong toolchain version, missing system dependencies, flaky network calls, etc. — would substantially strengthen the paper and guide future pipeline improvements.
4. Model evaluation at evaluation time conflates task difficulty with dataset issues. Table 6 shows pass rates ranging from 8.8% (gpt-oss-120b) to 25.2% (Claude Opus 4.5). While these numbers are useful baselines, they are not disambiguated from environmental noise. It is unclear how many failures are due to B-category issues vs. genuine model limitations.
5. Diagnostic study language coverage is narrow. The task analysis in Section 4.3 covers only five languages (Python, JavaScript, Go, Rust, Scala) — all relatively well-resourced. This research's important finding is that the failure mode taxonomy (B1–B7) was derived from this subset, yet it is applied globally to all 20 languages. Long-tail languages with heterogeneous toolchains may exhibit qualitatively different failure modes not captured by the current taxonomy.
6. Recent work has expanded evaluation beyond Python through multilingual benchmarks, such as Multi-SWEbench (Zan et al., 2025) and SWE-PolyBench (Rashid et al., 2025), demonstrating that language diversity materially changes agent behavior and task difficulty. At the same time, these datasets highlight a persistent tension: achieving high-confidence executable instances often requires substantial manual verification. Recent multi programming languages coding benchmark Multi-LCB (ICLR 2026) omits manual verification by reusing programming competition problems, the authors could clarify key differences with their approach.

---

> ### Author Rebuttal · Authors · 2026-03-31
>
> We thank the reviewer for the thoughtful and detailed feedback. We address the main concerns below.
>
> Regarding the release of datasets and pre-built artifacts, we would like to clarify that the supplementary materials already include the codebase together with data samples. The full-scale datasets and pre-built Docker artifacts were not attached to the submission because their size exceeds the supplementary material limits. We are fully committed to releasing all artifacts at full scale upon publication, and the final version will include links to the public resources: the datasets will be hosted on Hugging Face, the code on GitHub, and the pre-built images on DockerHub.
>
> We agree that downstream training validation is an important missing piece. As also noted in our responses to other reviewers, the most direct validation of a training-oriented dataset would be a downstream training experiment, for example comparing filtered and unfiltered subsets or evaluating curriculum strategies. This is also one of our main goals for future work, especially at multilingual scale. At the same time, we believe that a meaningful training run for this dataset would require substantial additional compute and infrastructure. Instead, in this work we aimed to establish that the dataset has the properties required for such training: the tasks are executable, non-trivial, and admit measurable headroom across repeated attempts. In addition, during the rebuttal period we prepared an ablation comparing performance on A-category versus B-category tasks to better separate model limitations from environment noise. We will include these results in the final version of the manuscript. The detailed results are attached to the response to the [Reviewer o4vH](https://openreview.net/forum?id=UCAda9kS57&noteId=b79Ws7CpjL).
>
> Regarding the setup synthesis success rate, we agree that a deeper analysis of failure modes would strengthen the paper. The failure patterns we observe at this stage are broadly similar to those seen in standard SWE agent trajectories: the model can commit early to an incorrect solution, fail to recover after an execution error, or stop refining after an initially plausible but incomplete setup. There are also setup-specific failure modes of the kind the reviewer mentions. We chose not to allocate more effort to this stage because we viewed later stages of the pipeline as more critical for the present paper. At the same time, we agree that this is an important direction for future improvement and we will expand the discussion of setup failure modes in the final version. We also note that Table 2 already shows a practical mitigation: increasing the number of attempts significantly improves the setup resolution rate, making Pass@k-style inference an effective strategy at this stage.
>
> We agree that model evaluation can conflate task difficulty with imperfections in the dataset or environment. To address exactly this concern, during the rebuttal period we prepared an additional ablation reporting solved rates separately for A-category and B-category tasks the numbers of which are attached above. This massive performance gap successfully disentangles the two factors: failures in the Code A split reflect genuine model limitations on solvable tasks, while the near-zero performance in the Code B split confirms the presence of environmental noise. Most importantly, this proves that our proposed metadata filtering successfully isolates such tasks.
>
> Regarding the diagnostic study language coverage, we agree that analyzing only a subset of languages is a limitation. The trajectory- and instance-level failure analysis required to derive the B1–B7 taxonomy is highly resource-intensive, so we began with the most widely represented and practically important languages. We do not claim that this taxonomy is necessarily exhaustive for all 20 languages, especially for long-tail languages with more heterogeneous ecosystems and toolchains. Expanding the analysis to broader language coverage is an important direction for future work, and we plan to continue updating the dataset and its diagnostics over time.
>
> Our work is not intended to compete with Multi-LCB or SWE-PolyBench as just another evaluation benchmark. Instead, we target a different need: providing large-scale, reproducible environments specifically for training software engineering agents. Unlike Multi-LCB, which uses isolated competitive programming problems with simple I/O, we focus on real-world agentic software engineering.

---

> > ### Author Rebuttal · Reviewer_EoAC · 2026-04-02
> >
> > The authors have chosen to leave most of my suggestions for future work, which is entirely understandable. I would like to maintain my current score.

---

### Official Review · Reviewer_RJVY · 2026-03-13

**Soundness:** 4
**Presentation:** 3
**Significance:** 3
**Originality:** 3
**Overall Recommendation:** 6
**Confidence:** 4

**Summary:**

This work presents a language-agnostic automated pipeline for constructing large-scale, executable software engineering tasks intended for RL training of SWE agents. The core deliverable is a dataset and the pipeline which automates environment setup via an interactive agent, validates tasks through dual-pass execution, filters by issue clarity using an LLM ensemble, and enriches each instance with diagnostic metadata.

**Compliance With Llm Reviewing Policy:**

Affirmed.

**Final Justification:**

I am upgrading my recommendation to a Strong Accept (Score: 6). The sheer scale, diversity, and practical utility of this contribution cannot be overstated. The authors have constructed a monumental, multilingual dataset (36,000+ containerized tasks across 20 languages) complete with pre-built Docker environments and fail-to-pass test oracles. While I initially had reservations about the lack of a downstream training experiment, the rebuttal successfully convinced me that the dataset's construction, validation, and the engineering effort involved are more than sufficient for a single, highly impactful publication. This work provides the exact infrastructure the open-source community desperately needs to advance the training of software engineering agents.

**Key Questions For Authors:**

1. What is the measured rate of solution leakage in PR-derived tasks, and how effective are your leakage detectors (precision/recall)?
2. How do you handle flaky tests and nondeterminism. Do you have quantified flakiness rates and mitigation (retries, quarantining)?
3. What explains the unusually low performance of gpt-oss-120b in Table 6, and does this affect the reliability of clarity filtering?
4. How stable are tasks over time (dependency drift, repo changes)? Do you plan periodic rebuilds or pinned snapshots?
5. What licensing/compliance checks did you apply when packaging repositories into containers for redistribution?

**Limitations:**

yes

**Strengths And Weaknesses:**

## Strengths

- The proposed dataset is large: more than 100,000 PR-derived tasks across 20 languages. That scale and diversity exceed most prior benchmarks. The long-tail language accommodations include relaxed filters and reusable base images.

- The paper includes strong ablations for a dataset release. It compares interactive vs. non-interactive setup across models and retry budgets (Table 2). It also studies issue-clarity filtering across prompts, models, and ensembling (Tables 3–5).

- The paper identifies recurring failures such as test-suite coupling, implicit naming requirements, and external dependencies. It then feeds these insights back into metadata enrichment.

- The paper shows issue linkage is a major bottleneck for scale. Using PR descriptions to synthesize problem statements is a pragmatic workaround. Shipping both issue-linked and PR-derived sets with separate metadata helps users choose trade-offs.

## Weaknesses

- The biggest gap is the lack of evidence that training on this dataset improves agents. For an RL training oriented benchmark, that is a major missing validation. Curriculum and metadata claims remain largely untested.

- Language agnostic still depends on per-language templates and engineering. The deeper analysis focuses on only six major languages (Table 7). Long-tail language quality and setup success are not quantified much.

- PR-derived tasks rely on LLM-written statements based on PR text and patches. The paper mentions leakage detection, but does not quantify leakage rates or statement fidelity. That leaves a large quality uncertainty for the bigger split.

---

> ### Author Rebuttal · Authors · 2026-03-31
>
> We thank the reviewer for the thoughtful and detailed feedback.
>
> We agree that the strongest validation of a training-oriented dataset would be a downstream training experiment showing agent improvement. This is also our long-term goal. However, a convincing RL run at this scale would require substantial additional compute and infrastructure. Instead, we aimed to validate that the dataset has the properties needed for such training: Table 6 shows that stronger models solve more tasks than weaker ones, and that Pass@k is consistently higher than Pass@1, indicating that tasks are solvable but non-trivial and that multiple attempts provide measurable headroom, i.e., the regime where group-based RL methods can benefit.
>
> For metadata-based filters, we also report performance separately on A and B* categories, which have similar distributions of language, num_modified_files, and num_modified_lines. This partially disentangles model limitations from failures caused by environment noise or task imperfections. The detailed results are attached to the response to the [Reviewer o4vH](https://openreview.net/forum?id=UCAda9kS57&noteId=b79Ws7CpjL).
>
> Regarding language-agnosticism, we expanded Table 7 to cover all languages; due to length limits we attach only a subset below, but will update the paper with the whole table. Table 7 reflects only the initial repository-filtering stage. For dependency installation and setup success, we reported detailed numbers only for languages with reliable ground truth from existing benchmarks such as SWE-bench Verified, SWE-bench Multilingual, and Multi-SWE-Bench. Because dependency installation avoids manual language-specific engineering (all templates for a new language are generated automatically without manual intervention), we believe the six-language analysis provides partial evidence for broader generalizability, though we agree long-tail language quality should be quantified more explicitly. We will clarify this in the final version and emphasize that, as shown in Table 2, setup success improves with Pass@k inference, making the approach practically appealing across all languages.
>
> **Table 7:**
>
> | language   | total_tasks | total_repos | kept_repos | kept_tasks | kept_tasks_share | kept_repos_share |
>
> | rust       | 49703       | 3325        | 530        | 39397      | 0.79             | 0.16             |
>
> | php        | 29130       | 2607        | 492        | 23181      | 0.80             | 0.19             |
>
> | kotlin     | 13003       | 913         | 913        | 13003      | 1.00             | 1.00             |
>
> Q1. We ran an internal validation study on SWE-bench Pro using original PRs. We constructed 509 PR-derived tasks where the model had to generate a plausible problem statement from the PR patch and description. Using an LLM-as-a-Judge, we compared the generated statements against the SWE-bench Pro reference fields (Problem Statement, Requirements, Interfaces) and checked whether they introduced implementation details beyond the reference. Under this setup, 392/509 tasks (77.0%) were clean (rated 0-1), 117/509 (23.0%) showed some leakage (rated 2-3), and 12/509 (~2.4%) contained explicit solution leakage (rated 3). We will include this analysis in the revised paper and highlight that PRs are an additional source of tasks of larger scale but with lower quality, which is important to keep in mind.
>
> Q2. We mitigate flakiness with retries: each task is run three times, and we keep only instances whose test outcomes are unchanged across runs. This filtered out about 3% of tasks, reducing the set from 33,049 to 32,079 instances.
>
> Q3. We do not interpret the lower score of gpt-oss-120b as meaning the model is inadequate in absolute terms; rather, in our benchmark it is outperformed by particularly strong models such as DeepSeek v3.2, GLM-4.7, and Claude Opus 4.5. We also note that agentic SWE is substantially harder than clarity filtering, which is a much narrower single-turn generation task. For clarity filtering, we optimized for precision and found gpt-oss-120b sufficiently good for issue filtering.
>
> Q4. Main dataset instances are packaged into Docker images, which makes them reproducible and largely protects them from repository evolution. Dependency drift can still occur, e.g., when installation depends on external development packages that later disappear or change. We therefore view ongoing maintenance as important and plan to keep improving the pipeline, expand environment coverage, and enrich metadata with additional insights, including from trajectories in more languages.
>
> Q5. The dataset is released under CC BY 4.0. Each instance also includes the license of the underlying repository at the corresponding commit, and users are expected to comply with that repository’s license. For packaging and redistribution, we restricted the dataset to repositories with permissive licenses only.

---

> > ### Author Rebuttal · Reviewer_RJVY · 2026-04-03
> >
> > Thank you to the authors for their thoughtful and comprehensive rebuttal. I appreciate the time and care taken to respond to the questions and concerns raised during the review process. After reading the author response, I find that my concerns have been adequately addressed, and the clarifications provided have significantly improved my understanding of the paper and my confidence in its overall quality.
> >
> > Accordingly, I am increasing my score.

---

### Official Review · Reviewer_si4Z · 2026-03-16

**Soundness:** 2
**Presentation:** 2
**Significance:** 3
**Originality:** 2
**Overall Recommendation:** 4
**Confidence:** 3

**Summary:**

SWE-rebench V2 is an automated pipeline for mining executable software engineering tasks from GitHub PRs at scale. It uses an interactive LLM agent to synthesize Docker environments, validates tasks via fail-to-pass test extraction, and filters underspecified instances with an LLM ensemble. The main release is 36K+ containerized tasks across 20 languages and 3,800+ repos, plus 100K+ PR-derived tasks with LLM-generated problem statements. The paper ablates setup synthesis and issue clarity filtering, and provides a diagnostic study identifying failure categories across 5 languages and 7 models.

**Compliance With Llm Reviewing Policy:**

Affirmed.

**Final Justification:**

After reading the author rebuttal, I have updated my score, and the paper has very good potential (though I still think some small-scale RL experiments would make the paper more robust).

**Key Questions For Authors:**

Do the authors conduct any downstream training experiments showing that training on SWE-rebench V2 improves agent performance on a test set? A positive result here would significantly strengthen the paper and could raise my score.

**Limitations:**

The authors discuss limitations including the absence of training ablations and the restriction to single-container environments. However, without a downstream training experiment, it is hard to evaluate whether the dataset and its filtering/metadata actually serve their stated purpose of enabling better SWE agent training.

**Strengths And Weaknesses:**

## Strengths

The paper addresses a timely bottleneck: the scarcity of large-scale, multilingual, executable training environments for RL-based SWE agents. 36K+ tasks across 20 languages and 3,800+ repos is a meaningful advance beyond prior work, which is either Python-only or limited to a handful of languages at evaluation scale.

## Weaknesses

The central weakness is the absence of any downstream training experiment. The paper's stated motivation is enabling RL training, yet no training result is presented. Without this, we cannot assess whether the dataset actually improves agent capabilities or whether the metadata-based filtering produces better curricula. The paper remains a resource description without validation of its stated purpose.

Additionally, total pipeline compute (API calls, Docker build hours, storage) is not reported. Including this information would helpful.

---

> ### Author Rebuttal · Authors · 2026-03-31
>
> We thank the reviewer for the thoughtful feedback. We address the main concerns below.
>
> We agree that an RL training experiment would provide the strongest validation of our claim that the dataset is useful for training. More broadly, one of our main long-term goals is to enable large-scale training of software engineering agents, including in the multilingual setting. At the same time, we omitted this experiment in the present paper, as a meaningful run at the scale enabled by this resource would require substantial additional compute and infrastructure investment. To still provide evidence for the dataset’s utility without introducing a partial or underpowered training result, we focused on empirical signals already present in our data (Table 6). First, stronger models solve substantially more tasks than weaker ones; for example, frontier models such as Opus 4.5 significantly outperform smaller or less capable open models. Second, all evaluated models show consistently higher Pass@3 than Pass@1. Together, these observations suggest that the dataset contains tasks that are neither trivial nor uniformly unsolved: they are solvable, exhibit meaningful variation in difficulty, and admit non-zero improvement from repeated attempts. This is precisely the regime required to provide a strong learning signal for group-based RL methods (such as GRPO).
>
> During the rebuttal period we additionally calculated the resolved rates across tasks with A metadata label and B* labels to illustrate that our enrichment helps to separate the tasks into different groups of quality. The detailed results are attached to the response to the [Reviewer o4vH](https://openreview.net/forum?id=UCAda9kS57&noteId=b79Ws7CpjL).
>
> We also agree that reporting the pipeline compute is important for transparency, reproducibility, and helping readers understand the practical cost of constructing the resource. We have collected these statistics and will include them in the final version of the paper. The dependency-installation agent processed 21,692 repositories. On average, the agent required 24.67 turns per repository, resulting in approximately 535,000 total API calls. Per trajectory (repository), the agent processed an average of ~214,000 input tokens and generated ~966 output tokens. If routed through a commercial API provider, this token volume would cost approximately 0.0873 USD per run or  ~1,900 USD total. After obtaining the setup instructions, we validated dependency installations at the task level. This stage involved 583,809 Docker build runs. With an average build time of 2.71 minutes per run, this required approximately 26,400 total job-hours of compute. The pipeline successfully produced 32,079 final task environments. The total storage footprint for all resulting Docker artifacts is 26.36 TiB (roughly 27 TB).

---

> > ### Author Rebuttal · Reviewer_si4Z · 2026-04-02
> >
> > I thank the authors for the detailed rebuttal and the compute cost breakdown. However, I still believe even a small scale RL training experiment comparing SWE-rebench V2 against existing resources like SWE-Gym or Multi-SWE-RL would be necessary to validate the dataset's effectiveness and would significantly strengthen the paper.

---

### Decision · Program_Chairs · 2026-04-30

**Decision:**

Accept (regular)

**Comment:**

SWE-rebench V2 delivers an impressive large-scale resource for SWE agent training, featuring 36K+ containerized tasks across 20 languages, alongside 100K+ supplementary PR-derived tasks.

Reviewers were uniformly positive, recognizing the exceptional engineering effort and the dataset’s immediate practical value. Initial concerns regarding moderate novelty and missing downstream training validation were appropriately addressed in the rebuttal. The authors supplied concrete details on flakiness filtering, leakage prevention, and licensing, which collectively validated the robustness of the pipeline.

I recommend acceptance based on the paper’s clear merit as an infrastructure contribution. The dataset’s scale and quality speak for themselves, making it a strong asset for the community. However, consistent with the reviewers’ assessments, the lack of algorithmic novelty and downstream validation confines this to a solid, standard acceptance rather than a candidate for oral presentation.